# Characteristics and six-month viral load suppression of clients presenting with advanced HIV disease in South Africa

Elizabeth Kachingwe[1]*, Nyasha Mutanda[1], Vinolia Ntjikelane[1], Mariet Benade[1,2,3], Musa Manganye[4], Lufuno Malala[4], Sydney Rosen[1,2], Mhairi Maskew[1]

1 Faculty of Health Sciences, Health Economics and Epidemiology Research Office, University of the Witwatersrand, Johannesburg, South Africa, 2 Department of Global Health, Boston University School of Public Health, Boston, Massachusetts, United States of America, 3 Amsterdam Institute for Global Health and Development, Amsterdam University Medical Center, Amsterdam, The Netherlands, 4 National Department of Health, HIV/AIDS Treatment, Care and Support, Pretoria, South Africa

* ekachingwe@heroza.org

## Abstract

Despite advances in antiretroviral therapy (ART), a notable proportion of individuals still present with advanced HIV disease (AHD) at treatment initiation, defined by CD4 < 200 cells/µL or WHO stage 3/4. This group experiences higher mortality and more opportunistic infections. While guidelines exist, they often do not address AHD clients specific needs early in treatment. Addressing these gaps could improve outcomes. Between September 2022 and June 2023, we surveyed a sequential sample of clients initiating ART or ≤6 months post-initiation at 18 primary healthcare facilities across three provinces. This observational cohort collected socio-demographic data, HIV care history, and service delivery preferences, and linked surveys to routine medical records and described client characteristics using descriptive statistics. Relative risks and risk differences compared outcomes between AHD and non-AHD clients. Primary outcomes were 6-month retention and viral load suppression. Of 1,098 clients (72% female, median age 33), 938 had CD4 or WHO staging at ART start. Of these, 29% (n = 275) had AHD, with a median CD4 of 108 cells/µL. AHD clients were more often male (44% vs. 21%), older (median age: 38 vs. 31 years), and sought care due to illness (63% vs. 33%). TB diagnosis (42% vs. 12%) and testing (76% vs. 67%) were higher. Service preferences and health resource use were similar. Retention at 6 months was similar (80% vs. 75%), but mortality was higher in AHD clients (1.0% vs. 0.2%). AHD clients had more low-level viremia (24% vs. 11%; RR = 2.27, 95% CI = 1.67–3.09) and slightly lower suppression (43% vs. 47%). AHD remains a key barrier to optimal ART outcomes. Clients with AHD experienced poorer viral suppression, despite similar retention rates highlighting the need for early detection, tailored clinical support, and strengthened monitoring. Updating ART guidelines

**Data availability statement:** Data generated by the study will be made publicly available in the Open Boston University repository (https://open.bu.edu/) after the PREFER study protocol has been closed (anticipated closure December 2026). Until then, data will remain under the supervision of the Boston University Medical Campus IRB and the University of the Witwatersrand Human Research Ethics Committee (HREC). Requests can be sent to the BUMC IRB at medirb@bu.edu. Data extracted from routine medical records are owned by the study sites and the South African National Department of Health and cannot be made publicly available by the authors.

**Funding:** Funding for the study was provided by the Gates Foundation through award INV-031690 to Boston University. The funder had no role in study design, data collection, analysis, or preparation of this manuscript.

**Competing interests:** The authors have declared that no competing interests exist.

to address AHD-specific needs is critical to improving outcomes in this high-risk population.

## Introduction

South Africa, with an estimated 7.7 million people living with HIV and 5.9 million on antiretroviral treatment (ART) as of 2023 [1], continues to face a serious challenge with advanced HIV disease (AHD), a major contributor to HIV-related illness and death [2]. Advanced HIV disease at treatment initiation or re-initiation, typically defined by a CD4 cell count below 200 cells/µL or the presence of WHO clinical stage 3 or 4 conditions, is still very common among people newly diagnosed with HIV and those returning to care after a treatment interruption [3,4]. Studies estimate that as many as one third of adults starting ART in South Africa have AHD, with even higher rates among those re-initiating ART [5]. AHD resulting from ART failure while on treatment also remains a concern but is not the focus of this study.

Starting ART with advanced HIV disease is associated with poor clinical outcomes, such as high mortality rates and loss to follow-up (LTFU) [3,4]. It also adds strain to the healthcare system, as affected individuals have a higher risk of severe opportunistic infections, such as tuberculosis (TB), cryptococcal meningitis, and pneumocystis pneumonia, which require complex and intensive treatment [4,6]. A study in two sub-Saharan Africa countries reported notably high mortality among those with AHD 29.6% (95% CI 25.4%–34.3%) in Uganda and 17.2% (95% CI, 13.5%–21.6%) in Kenya. Similarly, a South African study found a mortality rate of 32.5 per 100 person-years among individuals with AHD within the first six months of ART initiation, often due to immune reconstitution inflammatory syndrome (IRIS) or untreated opportunistic infections [6–8]. Achieving consistent medication adherence and viral load suppression, a primary goal of ART, is challenging in people with AHD, who often have more complex treatment regimens [9–10], high viral loads, and severely weakened immune systems. ART retention and adherence may be more challenging for people with AHD for three reasons. First, almost by definition, developing AHD suggests that the patient faces barriers to starting or remaining on treatment, relative to those who present for treatment without AHD. These barriers, which may involve transport, employment, fear of stigma, or many other considerations, may not go away after ART is (re-) started. Second, the complexity of treatment regimens and increased pill burden and visit requirements associated with managing the opportunistic infections (OIs) or comorbidities that help define AHD may contribute to treatment fatigue and missed ARV doses. And third, patients with OIs may face challenges in taking or swallowing antiretroviral (ARV) medications due to symptoms such as oral or oesophageal lesions or may experience poor drug absorption caused by gastrointestinal complications [7,8].

In 2017, the WHO issued guidelines for managing patients with AHD, recommending rapid ART initiation within seven days and, when possible, same-day initiation for those without contraindications [9]. In South Africa, AHD has been addressed

in the national ART guidelines [2] and is the focus of a new set of WHO guidelines released in October 2024 [11]. While the guidelines provide clear clinical recommendations for the conditions associated with AHD, they cannot be tailored to the non-clinical characteristics or behaviour of AHD clients, because little is known about these characteristics. Whether individuals initiating ART with AHD differ from other initiators at baseline and/or have different short-term patterns of care, and whether such differences should guide us to differentiate interventions, remains unclear, both in South Africa and elsewhere.

In this paper, we use data from a survey of clients on ART from 0-6 months in South Africa to compare the demographic and socioeconomic characteristics, HIV history, treatment outcomes, and resource utilization of individuals presenting with advanced HIV disease (AHD) and those without AHD. We also report 6-month retention and viral suppression rates stratified by AHD status. By examining these differences, we seek to inform strategies for earlier detection and more effective management of AHD. This study contributes evidence on early ART outcomes disaggregated by AHD status within routine programmatic settings and highlights the potential impact of baseline sociodemographic and clinical factors as well as client preference on early treatment outcomes.

## Methods

### Ethics statement

The PREFER study was approved by the Boston University Institutional Review Board (South Africa H-42726, May 20, 2022) and by the University of the Witwatersrand Human Research Ethics Committee (South Africa M220440, August 23, 2022). The protocol for South Africa was approved by Provincial Health Research Committees through the National Health Research Database for each study district (August 1, 2022 for West Rand; September 1, 2022 for King Cetshwayo and August 28, 2022 for Ehlanzeni). The study is registered with ClinicalTrials.gov (NCT05454839).

PREFER ("Preferences for services in a patient's first six months on antiretroviral therapy for HIV in South Africa") was a prospective, observational cohort study conducted in South Africa and Zambia [10]. Its main goal was to inform the design of differentiated service delivery models for the early HIV treatment period, generally defined as the first six months after treatment initiation or re-initiation [12]. As PREFER enrolled a sequential sample of adults starting or re-starting ART at public sector clinics, some participants had AHD-defining conditions at study enrolment. We used survey data and electronic medical records to compare their characteristics and outcomes to non-AHD PREFER participants in South Africa.

### Study population and data collection

The study enrolled clients in 18 healthcare facilities in Gauteng, Mpumalanga and KwaZulu-Natal provinces of South Africa from September 2022 to June 2023. Clients eligible for enrolment in PREFER were adults (≥18 years) who had been on ART for ≤6 months; presented at study sites for ART initiation or re-initiation, routine HIV care, or unscheduled HIV-related care; and provided written informed consent. Individuals unable to communicate in the survey languages, unwilling to complete the survey on the day of consent, or deemed too ill to participate were excluded. Participants were recruited consecutively as they arrived at the facility, subject to interviewer availability. (In the remainder of this manuscript, we use "ART initiation" to refer to starting ART for both naïve and experienced clients (re-initiators).

The PREFER survey was a structured, interviewer-administered questionnaire that all participants completed at study enrolment. The survey collected data on client demographic and socioeconomic characteristics, HIV testing and treatment history, and preferences and expectations for service delivery during the first 6 months on ART. Using identifiers from the structured questionnaire, follow-up data from routinely collected medical records were accessed for each participant for up to 12 months after study enrolment (from September 2022 to June 2024). Data sources included the national client electronic medical register (Tier.Net), paper records and registers from participating facilities, and laboratory results from the National Health Laboratory Services (NHLS) database. We were unable to access laboratory results for participants

enrolled in the PREFER cohort in Zambia, and therefore could not define advanced HIV disease (AHD) using the same criteria applied to South African participants.

## Study variables and statistical analysis

The primary exposure variable for this study was AHD status at study enrolment. Client AHD status was ascertained using CD4 cell count test result and/or WHO stage at ART initiation. A client was classified as presenting with AHD if their CD4 count result closest to ART start was < 200 cells/ μL or they had a WHO stage 3 or 4 defining condition recorded at ART initiation. CD4 count results included tests conducted from six months before to one week after starting ART treatment.

We defined two primary outcome variables for this analysis: 1) Retention in care at 6 months after ART start; and 2) HIV viral load suppression. A client was considered retained in care if they attended all their scheduled clinic visits at the originating facility within 28 days of the scheduled visit date during the first 6 months on ART or had a documented transfer to another facility within 6 months of ART initiation. We included viral load measurements taken between three and nine months after ART initiation to estimate the 6-month viral suppression outcome. VL results were categorized as follows: suppressed (<50 copies/mL); low-level viremia (50–1000 copies/mL); or unsuppressed (≥1000 copies/mL). In line with the South African National Department of Health adherence guidelines, HIV viral load suppression categories for individuals on ART are defined as follows: undetectable (viral load <50 copies/mL), suppressed (detectable but ≤1,000 copies/mL), and unsuppressed (>1,000 copies/mL). For the purposes of this analysis, we adopted these thresholds but refer to the categories as suppressed, low-level viremia, and unsuppressed [13].

We defined re-initiation of ART as self-reported re-engagement in HIV care following an interruption in ART for a period of at least 90 days. This threshold was selected based on existing literature and programmatic definitions of disengagement from HIV care, which commonly use a 90-day threshold to distinguish between temporary non-adherence and more sustained treatment interruption. No maximum time period was specified. Participants who reported being out of care for less than 90 days were not classified as having re-initiated ART, as they were not considered to have disengaged from care per our study definition [14].

For the analysis, we first used frequencies and simple proportions for categorical variables and medians and inter-quartile ranges for continuous variables to compare client characteristics at study enrolment stratified by AHD status. The association between AHD status and the study's primary outcomes (6-month retention in care and viral suppression) were estimated using risk differences (RD) and risk ratios (RR) and are reported with 95% confidence intervals (CI). We then conducted a comparison of healthcare resource utilization between AHD and non-AHD clients. The comparison captured several services, including the number of clinic visits during the first six months of ART treatment, the proportion of clients receiving TB Preventive Therapy (TPT), the proportion on Cotrimoxazole Preventive Therapy (CPT), and the proportion of clients who had a six-month viral load (VL) test. We report baseline characteristics, patient HIV history, service delivery preferences, healthcare resource utilisation and six-months outcome stratified by AHD status.

## Results

### Client characteristics

Of the 1,098 individuals enrolled in the PREFER survey, 938 had a documented baseline CD4count, and 900 had WHO stage recorded at initiation, all clients without baseline CD4 count or WHO staging recorded at baseline were excluded from further analysis. Similar socio-demographic characteristics were observed between those with baseline CD4 counts or recorded WHO staging and those without (S1 Table). Of the remaining 938, 275 (29%) presented with advanced HIV disease (AHD) as defined by CD4 count or WHO stage. Of these, 259 (94%) had CD4 < 200 cells/ μL only, 41 (4%) presented with a WHO stage 3/4 defining condition only, and 25 (3%) met both criteria. AHD clients were older (median age 38 vs. 31 years), more often male (44% vs. 21%), and had higher rates of active or prior tuberculosis (42% vs 12%) than did non-AHD clients. (Table 1).

**Table 1. Description of the analytic cohort by AHD status.**

| Characteristic | Level | Total | Non-AHD | AHD |
|---|---|---|---|---|
| N (%) | | 938 | 663 (71) | 275 (29) |
| Age | Median age (IQR) | 33 (27-40) | 31 (25-38) | 38 (31- 44) |
| | 18-24 years | 169 (18) | 150 (23) | 19 (7) |
| | 25-49 years | 677 (72) | 463 (70) | 214 (78) |
| | 50+ years | 92 (10) | 50 (8) | 42 (15) |
| Sex | Female | 677 (72) | 523 (79) | 154 (56) |
| Marital status | Married-living with partner | 289 (31) | 199 (30) | 90 (33) |
| | Married-not living with partner | 441 (47) | 337 (51) | 104 (38) |
| | Single | 208 (22) | 127 (19) | 81 (29) |
| Education | Primary or less | 347 (37) | 238 (36) | 109 (40) |
| | Secondary | 453 (48) | 324 (49) | 129 (47) |
| | Post-secondary | 138 (15) | 101 (15) | 37 (13) |
| Occupation | Formal | 204 (22) | 132 (20) | 72 (26) |
| | Informal | 187 (20) | 126 (19) | 61 (22) |
| | Unemployed | 477 (51) | 347 (52) | 130 (47) |
| | Student/Trainee | 70 (7) | 58 (9) | 12 (4) |
| Food scarcity | Never | 658 (70) | 468 (71) | 190 (69) |
| | Seldom | 59 (6) | 38 (6) | 21 (8) |
| | Sometimes | 191 (20) | 135 (20) | 56 (20) |
| | Often | 30 (3) | 22 (3) | 8 (3) |
| Access to money for health care needs | Very difficult | 127 (14) | 89 (13) | 38 (14) |
| | Difficult | 392 (42) | 285 (43) | 107 (39) |
| | Easy | 378 (40) | 263 (40) | 115 (42) |
| | Very easy | 41 (4) | 26 (4) | 15 (5) |
| Facility total remaining on ART (TROA[1]) | <2000 | 236 (25) | 150 (23) | 86 (31) |
| | 2000-4000 | 371 (40) | 253 (38) | 118 (43) |
| | >4000 | 331 (35) | 260 (39) | 71 (26) |
| Patient ART timing | Initiating today | 273 (29) | 199 (30) | 74 (27) |
| | Re-engagement[2] | 135 (14) | 95 (14) | 40 (15) |
| | On treatment | 530 (57) | 369 (56) | 161 (59) |
| Baseline regimen | Tenofovir/Lamivudine/Dolutegravir | 586 (91) | 242 (89) | 828 (90) |
| | Tenofovir/Emtricitabine/Efavirenz | 47 (7) | 26 (10) | 73 (8) |
| | Other | 12 (2) | 4 (1) | 16 (2) |

[1] TROA: **(Total Remaining on ART)** refers to the total number of patients actively receiving antiretroviral therapy (ART) at a specific healthcare facility

[2] Re-engement in care refers to the process of returning to HIV treatment services after a period of **interruption or disengagement** from care (self-report)

Table 2 summarizes the HIV care history of participants stratified by AHD status at ART initiation. Nearly two thirds (63%) of AHD clients sought healthcare services due to illness, compared to 38% of non-AHD clients. A higher proportion of AHD clients than non-AHD clients reported experiencing health concerns after ART initiation (36% vs. 17%). AHD clients were less likely to initiate ART on the day of testing (64% vs. 82%) and more likely to attend monthly clinic visits for additional care services during the first 6 months on treatment (87% vs. 64%) than their non-AHD counterparts. They were also more likely to undergo tuberculosis testing before ART initiation (76% vs. 67%) and had a higher prevalence of TB diagnosis (42% vs.12%). Though the proportion of clients reporting ever previously interrupting treatment was the same (11%) for both groups, clients presenting

**Table 2. HIV care history by AHD at treatment initiation.**

| Characteristic | Level | Non-AHD | AHD | Total |
|---|---|---|---|---|
| N (%) | | 663 (71) | 275 (28) | 938 |
| Tested HIV positive before most recent ART initiation (yes) | Yes | 226 (34) | 90 (33) | 316 (34) |
| Reason for testing | Recommended by healthcare provider | 162 (24) | 43 (16) | 205 (22) |
| | Known exposure or risk | 88 (13) | 22 (8) | 110 (12) |
| | Ill health | 249 (38) | 172 (63) | 421 (45) |
| | Other | 20 (3) | 2 (1) | 22 (2) |
| | Pregnancy/antenatal | 41 (6) | 10 (4) | 51 (5) |
| | Self-initiated/Voluntary testing | 103 (16) | 26 (9) | 129 (14) |
| Treatment initiation | Same day | 541 (82) | 176 (64) | 717 (76) |
| | Within a week | 78 (12) | 73 (27) | 151 (16) |
| | Month plus | 40 (6) | 22 (8) | 62 (7) |
| | Don't know/can't remember | 4 (1) | 4 (1) | 8 (1) |
| Previous treatment (yes) | Yes | 73 (11) | 30 (11) | 103 (11) |
| Number of times client defaulted treatment in the past | Once | 67 (92) | 22 (73) | 89 (86) |
| | Twice | 2 (3) | 5 (17) | 7 (7) |
| | Three or more | 4 (5) | 3 (10) | 7 (7) |
| Client experienced illness related to ART (diarrhoea, nausea and fatigue) | Yes | 111 (17) | 100 (36) | 211 (22) |
| Client physical condition at the time of testing | I felt very sick | 74 (11) | 62 (23) | 136 (14) |
| | I felt a little sick | 165 (25) | 102 (37) | 267 (28) |
| | I felt fine, not sick at all | 424 (64) | 111 (40) | 535 (57) |
| Client physical condition today | I feel very sick | 7 (2) | 5 (3) | 12 (2) |
| | I feel a little sick | 22 (6) | 24 (15) | 46 (8) |
| | I feel fine, not sick at all | 358 (93) | 133 (82) | 491 (89) |
| Dispensing interval | None | 22 (3) | 8 (3) | 30 (3) |
| | 1 month | 546 (82) | 228 (83) | 774 (83) |
| | 2 months | 63 (10) | 28 (10) | 91 (10) |
| | 3 months | 28 (4) | 8 (3) | 36 (4) |
| | Other (specify) | 4 (1) | 3 (1) | 7 (1) |
| Client has missed scheduled facility visits by 2–3 days | Yes | 56 (8) | 12 (4) | 68 (7) |
| | Don't know/can't remember | 1 (0) | 2 (1) | 3 (0) |
| Number of people client has disclosed to | None | 168 (25) | 58 (21) | 226 (24) |
| | One person | 199 (30) | 93 (34) | 292 (31) |
| | Two people | 118 (18) | 46 (17) | 164 (17) |
| | Three people | 77 (12) | 31 (11) | 108 (12) |
| | Four people or more | 101 (15) | 47 (17) | 148 (16) |
| Number of people client believes to be aware of client's HIV status | No one | 299 (45) | 104 (38) | 403 (43) |
| | Just 1 or 2 others | 192 (29) | 92 (33) | 284 (30) |
| | 3-5 others | 94 (14) | 36 (13) | 130 (14) |
| | Over 6 others | 45 (7) | 23 (8) | 68 (7) |
| | Don't know | 33 (5) | 20 (7) | 53 (6) |
| Other person living in client's household know client's HIV status | Yes | 384 (58) | 168 (61) | 552 (59) |
| | No | 254 (38) | 82 (30) | 336 (36) |
| | Live alone | 25 (4) | 25 (9) | 50 (5) |

*(Continued)*

**Table 2.** (Continued)

| Characteristic | Level | Non-AHD | AHD | Total |
|---|---|---|---|---|
| Client knows person who has been taking ART for at least one year and is doing well | Yes | 540 (81) | 233 (85) | 773 (82) |
| Client's partner or husband or wife knows client has HIV | Yes | 347 (65) | 136 (70) | 483 (66) |
| | No partner | 13 (2) | 3 (2) | 16 (2) |
| Comorbidities (At study enrolment) | Tuberculosis | 6 (12) | 13 (42) | 19 (23) |
| | Diabetes | 6 (12) | 6 (19) | 12 (15) |
| | Hypertension | 28 (56) | 13 (42) | 41 (51) |
| | Asthma | 10 (20) | 1 (3) | 11 (14) |
| | Other | 4 (8) | 2 (6) | 6 (7) |
| Client comes to this clinic for services | Yes | 39 (78) | 27 (90) | 66 (82) |
| How often client comes to this clinic for additional chronic services other than HIV care | Monthly | 32 (64) | 26 (87) | 58 (72) |
| | Every 2 months | 6 (12) | 2 (7) | 8 (10) |
| | Every 3 months | 3 (6) | 0 (0) | 3 (4) |
| | Every 6 months | 0 (0) | 1 (3) | 1 (1) |
| | Other (specify) | 9 (18) | 1 (3) | 10 (12) |
| Client was asked about a cough | Yes | 585 (88) | 250 (91) | 835 (89) |
| Client had a TB test when starting HIV treatment most recently | Yes | 442 (67) | 208 (76) | 650 (69) |
| Client was told to wait to start HIV treatment until receiving TB results | Yes | 97 (22) | 74 (36) | 171 (26) |
| Client has ever been diagnosed with TB | Yes | 26 (4) | 39 (14) | 65 (7) |
| Of the clients ever been diagnosed with TB, number treated. | Yes | 21 (81) | 23 (59) | 44 (68) |
| TB treatment start | In the past 1 month | 3 (14) | 7 (30) | 10 (23) |
| | In the 6 months | 2 (10) | 4 (17) | 6 (14) |
| | More than 6 months ago | 16 (76) | 12 (52) | 28 (64) |
| TB treatment completion (of those who started treatment more than 6 months ago) | Yes | 16 (100) | 12 (100) | 28 (100) |

with AHD reported more frequent prior interruptions than the non-AHD group, with 27% of the AHD group reporting two or more interruptions compared to 8% of the non-AHD group. Both groups reported a similar proportion of pregnancies (2% vs. 3%)

## Retention in care and viral suppression at 6 months post ART initiation

At 6 months after ART initiation, AHD clients experienced similar or slightly better rates of retention in care than their non-AHD counterparts (Table 3). In total, 80% of AHD clients were observed in care at 6 months, compared to 75% of non-AHD clients (RR = 1.06; 95% CI: 0.99–1.14). AHD clients had higher rates of mortality than non-AHD clients (1% vs. 0.2%), however, though few deaths were observed in either group. No differences in the proportion interrupting treatment and returning to care were noted between groups.

AHD clients were somewhat more likely to have a VL test result observed at 6 months after initiation (28% those with AHD were missing a VL test vs 36% among those without AHD) and, importantly clients with AHD were more than twice as likely to experience low-level viremia (24% vs. 11%; RR = 2.27 95% CI 1.67-3.09) compared to non-AHD clients. The AHD cohort was also somewhat less likely to achieve viral suppression (43% vs. 47%; RR = 0.93; 95% CI 0.80-1.09) by 6 months on ART. Among AHD clients with low-level viremia, 91% were on a Dolutegravir-based regimen, compared to 87% of the non-AHD cohort with low-level viremia.

**Table 3. Clinical outcomes at 6 months post ART initiation stratified by AHD status.**

| Characteristic | Level | Non-AHD (n = 663; 71%) | AHD (n = 275; 29%) | Risk difference (95% CI) | Relative risk (95% CI) |
|---|---|---|---|---|---|
| **RETENTION IN CARE BY 6 MONTHS ON ART** | | | | | |
| Continuously in care | Overall | 497 (75%) | 219 (80%) | 5% (-1–10%) | 1.06 (0.99-1.14) |
| | *In care at initiating facility* | *466 (70%)* | *208 (75%)* | *5% (-1–12%)* | *1.08 (0.99-1.17)* |
| | *Transferred to another facility* | *31 (5%)* | *11 (5%)* | *0% (-3–2%)* | *0.86 (0.44-1.68)* |
| Disengaged from care | Overall | 148 (22%) | 53 (19%) | -3% (-9–3%) | 0.86 (0.65-1.14) |
| | *Disengaged immediately after initiation* | *21 (3%)* | *3 (1%)* | *-2% (-4–0%)* | *0.34 (0.10-1.15)* |
| | *Disengaged from care 1–6 months after initiation* | *37 (6%)* | *14 (5%)* | *0% (-4–3%)* | *0.91 (0.50-1.66)* |
| | *Deceased* | *1 (0.2%)* | *3 (1%)* | *–* | *–* |
| | *Interrupted treatment and returned to care* | *89 (13%)* | *33 (12%)* | *-1% (-6–3%)* | *0.89 (0.62-1.30)* |
| Outcome unknown | | 18 (3%) | 3 (1%) | -2% (-3–0%) | 0.40 (0.12-1.35) |
| **VL SUPPRESSION BY 6 MONTHS ON ART (COPIES/ML)** | | | | | |
| VL suppressed (<50 copies/mL) | | 314 (47%) | 121 (43%) | -4% (-10–4%) | 0.93 (0.80-1.09) |
| Low-level viremia (50–999) | | 70 (11%) | 66 (24%) | 13% (8–19%) | 2.27 (1.67-3.09) |
| Unsuppressed VL (≥1000 copies/mL) | | 43 (6%) | 11 (4%) | -2% (-5–0%) | 0.62 (0.32-1.18) |
| VL result not observed at 6 months | | 236 (36%) | 77 (28%) | -8% (-14 to -2%) | 0.78 (0.63-0.96) |

## Preferences for service delivery

AHD clients expressed preferences for service delivery options that were largely similar to those of non-AHD clients (Table 4). Similar proportions of AHD and non-AHD clients preferred bimonthly clinic visits (28% vs. 24%), mid-month appointments (19% vs. 15%), and attending clinic visits alone (88% vs. 85%). A comparable proportion of both groups preferred their medication packaging in an unmarked (blank) container (10% vs. 6%) and to be seen by a doctor (17% vs. 13%). Preferences for written treatment literacy materials (42% vs. 34%) and one-on-one counselling sessions (52% vs. 48%) were also similar between the two groups. Symptomatic AHD clients (diagnosed by WHO stage 3/4 and CD4 count <200 cells/µ L) and those diagnosed with CD4 count <200 cells/µL alone reported differences in service delivery expectations and preferences. Clients with CD4 < 200 cells/µL were more likely to prefer the clinic every month (22% vs. 15%) and prefer receiving medication in 1-month intervals (20% vs. 12%). They also preferred their visits to be early in the month (41% vs. 22%). In contrast, symptomatic clients (WHO stage 3/4 and CD4 < 200 cells/µL) were more likely to prefer external pick-up services (67% vs. 51%) (S2 Table).

## Resource utilization

Among clients retained in care for at least six months (n = 716), similar proportions of AHD (67%) and non-AHD (63%) clients attended six or more clinic visits during the first six months of ART (RD: 3%, 95% CI: -4% to 10%, RR: 1.0, 95% CI:

**Table 4. Clients' preferences stratified by AHD status.**

| Preference | Level | Non-AHD | AHD | Total |
|---|---|---|---|---|
| N (%) | | 663 (71) | 275 (29) | 938 |
| Offered choices? | Yes | 41 (6) | 16 (6) | 57 (6) |
| Frequency of clinic visit | Every month | 86 (13) | 44 (16) | 130 (14) |
| | Every 2 months | 162 (24) | 77 (28) | 239 (25) |
| | Every 3 months | 281 (42) | 110 (40) | 391 (42) |
| | Every 6 months | 123 (19) | 42 (15) | 165 (18) |
| | Other (specify) | 11 (2) | 2 (1) | 13 (1) |
| Dispensing intervals | 1 month at a time | 79 (12) | 37 (13) | 116 (12) |
| | 2 months at a time | 160 (24) | 78 (28) | 238 (25) |
| | 3 months at a time | 286 (43) | 115 (42) | 401 (43) |
| | 4 months at a time | 12 (2) | 3 (1) | 15 (2) |
| | 6 months at a time | 126 (19) | 42 (15) | 168 (18) |
| Part of the month | Early in the month (first week) | 219 (33) | 69 (25) | 288 (31) |
| | Late in the month (last week) | 95 (14) | 44 (16) | 139 (15) |
| | Middle of the month | 97 (15) | 51 (19) | 148 (16) |
| | Doesn't matter, can come any time during the month | 252 (38) | 111 (40) | 363 (39) |
| Day of the week | Monday | 246 (37) | 91 (33) | 337 (36) |
| | Tuesday | 231 (35) | 90 (33) | 321 (34) |
| | Wednesday | 262 (40) | 116 (42) | 378 (40) |
| | Thursday | 242 (37) | 94 (34) | 336 (36) |
| | Friday | 262 (40) | 104 (38) | 366 (39) |
| | Saturday | 170 (26) | 60 (22) | 230 (25) |
| | Sunday | 129 (19) | 43 (16) | 172 (18) |
| Time of the day | Before work in the morning (before 8 am) | 230 (35) | 103 (37) | 333 (36) |
| | Mornings (8 am to 12 pm) | 367 (55) | 144 (52) | 511 (54) |
| | Lunch time (12 am to 2 pm) | 64 (10) | 20 (7) | 84 (9) |
| | Afternoons (2–4 pm) | 71 (11) | 18 (7) | 89 (9) |
| | After work in the early evening (4–7 pm) | 22 (3) | 9 (3) | 31 (3) |
| | Other | 40 (6) | 16 (6) | 56 (6) |
| Accompanied to the facility | Alone | 562 (85) | 241 (88) | 803 (86) |
| | With a family member | 101 (15) | 34 (12) | 135 (14) |
| External pick up | Yes | 431 (65) | 178 (65) | 609 (65) |
| Home delivery | Yes | 343 (52) | 148 (54) | 491 (52) |
| Medication packaging | One bottle for each month | 290 (44) | 117 (43) | 407 (43) |
| | One larger bottle with several months in it | 98 (15) | 50 (18) | 148 (16) |
| | An unmarked (blank) container | 42 (6) | 27 (10) | 69 (7) |
| | A container with instructions on it | 41 (6) | 21 (8) | 62 (7) |
| | A blister pack | 102 (15) | 24 (9) | 126 (13) |
| | Any kind of packaging is fine | 89 (13) | 36 (13) | 125 (13) |
| | Something else(specify) | 1 (0) | 0 (0) | 1 (0) |
| Provider choice | Doctor or clinical officer | 86 (13) | 47 (17) | 133 (14) |
| | Nurse | 534 (81) | 214 (78) | 748 (80) |
| | Counsellor | 35 (5) | 10 (4) | 45 (5) |
| | Other | 8 (1) | 4 (1) | 12 (1) |

*(Continued)*

| Preference | Level | Non-AHD | AHD | Total |
|---|---|---|---|---|
| More information | More | 312 (47) | 138 (50) | 450 (48) |
| | The same | 334 (50) | 125 (45) | 459 (49) |
| | Less | 17 (3) | 12 (4) | 29 (3) |
| More counselling | More | 320 (48) | 141 (51) | 461 (49) |
| | The same | 331 (50) | 125 (45) | 456 (49) |
| | Less | 12 (2) | 9 (3) | 21 (2) |
| Information format | Written material (brochure or information sheet) | 227 (34) | 115 (42) | 342 (36) |
| | Class/group session in community (not at clinic) | 58 (9) | 19 (7) | 77 (8) |
| | Class/group session with provider at clinic | 111 (17) | 46 (17) | 157 (17) |
| | One-on-one session with provider at clinic | 321 (48) | 143 (52) | 464 (49) |
| | Social media (e.g., Facebook, Twitter) | 159 (24) | 47 (17) | 206 (22) |
| | Community group in my community | 27 (4) | 8 (3) | 35 (4) |
| | Radio or TV | 153 (23) | 67 (24) | 220 (23) |
| | Videos I can watch online at home | 76 (11) | 23 (8) | 99 (11) |
| | Text messages on my phone | 376 (57) | 139 (51) | 515 (55) |
| | Links to websites that I can browse in my own time | 119 (18) | 49 (18) | 168 (18) |
| | Other specify | 1 (0) | 2 (1) | 3 (0) |
| Facility care | As good as expected | 524 (79) | 201 (73) | 725 (77) |
| | Better than expected | 119 (18) | 63 (23) | 182 (19) |
| | Worse than expected | 20 (3) | 11 (4) | 31 (3) |

0.9 to 1.2). The proportions receiving TPT were the same between groups (78%) (Risk Difference: 0.7%, 95% CI: -6% to 7%, RR: 1.0, 95% CI: 0.9 to 1.1). Similarly, 77% of AHD patients and 73% of non-AHD patients had their viral load documented at six months (RD: 4%, 95% CI: -2% to 10%, RR: 1.1, 95% CI: 1.0 to 1.2). Among the group with AHD, females were more likely to have documented 6 months viral load compared to males (77 vs.65%). A high proportion of females (44%) preferred early morning clinic visits before 8 am, compared to males (30%). Additionally, males (59%) had a stronger preference for one-on-one counselling sessions with a provider at the clinic than females (47%). When it comes to facility care, a higher percentage of males (79%) rated the care as "as good as expected" compared to females (69%), with more females (27%) rating the service as "better than expected". (S3 and S4 Tables).

Symptomatic clients (diagnosed by WHO stage 3/4 and CD4 count <200 cells/µL) and those diagnosed with CD4 count <200 cells/µL alone reported differences in service utilization and service delivery expectations and preferences (Table 5). Clients with CD4 < 200 cells/µL alone were slightly more likely to prefer the clinic every month (22% vs. 15%) and prefer receiving medication in 1-month intervals (20% vs. 12%). They also preferred their visits to be early in the month (41% vs. 22%). In contrast, symptomatic clients (WHO stage 3/4 and CD4 < 200 cells/µL) were more likely to prefer external pick-up services (67% vs. 51%). A slightly higher proportion of CD4 < 200 cells/µL clients (93% vs. 80%) were not receiving cotrimoxazole therapy than were symptomatic clients (S5 Table).

## Discussion

In this analysis of HIV treatment clients initiating or reinitiating care across three provinces in South Africa in 2022–2023, 29% of study participants presented with advanced HIV disease, a proportion consistent with national estimates [5]. We found that participants with advanced HIV disease were as likely as those without AHD to remain in care 6 months later but were twice as likely to experience low-level viremia and somewhat less likely to achieve viral suppression by 6 months

**Table 5. Healthcare resource use among AHD and non-AHD clients.**

| Variable | Level | Non-AHD (n = 497; 71%) | AHD (n = 219; 29%) | Risk difference (95% CI) | Relative risk (95% CI) |
|---|---|---|---|---|---|
| Number of clinic visits in the first 6 months of ART among clients continuously in care | ≥6 visits | 315 (63) | 145 (67) | 3% (-5% to 10%) | 1.04 (0.93 to 1.17) |
| | <6 visits | 182 (37) | 74 (33) | -3% (-10% to 5%) | 0.90 (0.74 to 1.15) |
| | Median (IQR) | 5 (4-6) | 5 (4-6) | | |
| TPT | No | 110 (22) | 47 (22) | -0.7% (-6% to 7%) | 1.00 (0.80 to 1.41) |
| | Yes | 387 (78) | 172 (78) | 0.7% (-6% to 7%) | 1.0 (0.93 to 1.11) |
| Cotrimoxazole preventive therapy | No | 456 (92) | 191 (87) | -4% (9% to 1%) | 1.00 (0.90 to 1.01) |
| | Yes | 29 (6) | 18 (8) | 2% (-1.8%% to 6%%) | 1.41 (0.80 to 2.48) |
| | Unknown | 12 (2) | 10 (5) | 2% (-1–5%) | 1.89 (0.83 to 4.31) |
| Six-month viral load documented | No | 136 (27) | 51 (23) | -4% (-1% to 0.14) | 0.85 (0.64 to 1.13) |
| | Yes | 361 (73) | 168 (77) | 4% (-2% to 10%) | 1.06 (0.96 to 1.16) |

on ART. Our results indicate that AHD clients were older, more likely to be male, and more likely to seek care due to illness compared to non-AHD clients. These findings mirror other studies, which have shown that delayed HIV diagnosis and care engagement are more common among men and older adults [3,4]

Other studies have reported inconsistent relationships between AHD and clinical outcomes (retention in care, viral suppression), such that our findings agree with some previous work and disagree with other. Similar to our results, an earlier South African study showed no link between lower CD4 count at ART initiation and loss to follow-up (LTF), after adjusting for unascertained deaths [4]. A multi-country study elsewhere in sub-Saharan Africa, however, reported that AHD was associated with a combined LTF and death endpoint; while clients with CD4 counts of 100–199 cells/ μL had similar retention outcomes to those with CD4+ ≥ 200 cells/ μL, clients with CD4 + counts <100 cells/ μL were at significantly higher risk for poor outcomes [6]

In our study, AHD clients exhibited a higher likelihood of low-level viremia at six months, with a twofold increased relative risk compared to non-AHD clients. This aligns with previous studies demonstrating that AHD is associated with delayed viral suppression due to higher baseline viral loads and greater immunological deficits. People with AHD face greater challenges in achieving suppression due to complex treatment regimens, high viral loads, and severely weakened immune systems [15,16]. Factors such as immune dysfunction, incomplete immune recovery, and poor adherence to ART, often exacerbated by comorbidities, lead to partial viral suppression and contribute to the persistence of low-grade viremia [17]. Low-level viremia in turn can lead to chronic immune activation and inflammation, which may contribute to non-AIDS-related comorbidities, such as cardiovascular disease, kidney disease, and cognitive decline. Low-level viremia was reported as an independent risk factor of virologic failure ([18,19]. Managing low-level viremia (LLV) comes with both clinical and policy implications. On the clinical side, it may indicate that patients need closer monitoring, extra support to stick to their treatment, and/or a change in their medication to avoid drug resistance and treatment failure. From a policy perspective, clear guidelines are needed to decide when to step in, along with enough resources for extra testing and

support programs. It is also important to improve data systems like TIER.Net so that cases of low-level viremia are properly tracked and to ensure that healthcare workers are trained to recognize and manage it. Addressing these issues is key to helping patients to achieve favourable treatment outcomes and keeping HIV treatment effective overall.

We observed that timing of ART initiation varied between AHD and non-AHD clients, with non-AHD clients significantly more likely to start treatment on the same day than were AHD clients. By the end of one week, though, this difference had largely disappeared, with 91% of AHD clients and 94% of non-AHD clients on ART by the 7-day mark. Although retention in care did not vary between the groups, as noted above, both groups experienced high loss to follow up by six months, 22% for those without AHD, 19% for those with, confirming the conclusion that retention in care after initiation has become the most pressing challenge for national ART programs, unlike in previous periods [20,21].

Tuberculosis and hypertension were the most common co-morbidities among AHD clients in our study; those without AHD were more likely to report hypertension (56% vs, 46%) but much less likely to have TB. The high prevalence of TB among AHD clients is not surprising, as TB is itself a criterion for AHD. More surprising is that a substantially larger proportion of non-AHD clients reported prior TB treatment (81% vs. 59%), though AHD clients reported more recent TB treatment. These results suggest that previous, treated TB is not a risk factor for current AHD, a finding that should be explored further.

While tuberculosis was more prevalent among AHD clients, hypertension was more frequently reported among clients without AHD. This finding aligns with known immunological and metabolic consequences of HIV infection at different stages of disease progression. Higher rates of hypertension among clients without AHD may be attributable to chronic immune activation and systemic inflammation associated with long-term HIV infection and antiretroviral therapy (ART), even in the absence of advanced disease [22].

These findings highlight the importance of integrated and differentiated care models. While TB screening and treatment remain central to care for clients with AHD, the high burden of hypertension among those without AHD highlights the need for routine screening and management of non-communicable diseases (NCDs) within HIV care programs. Integrated models that address both communicable and non-communicable comorbidities have been shown to improve outcomes and are increasingly relevant as HIV populations age and live longer on ART [23,24].

Service delivery preferences of clients with advanced HIV disease (AHD) were largely similar to those of non-AHD clients, with only minor differences observed. Our study also found little to no difference in healthcare utilization between AHD and non-AHD clients during the first six months of ART. A similar proportion of AHD clients (67%) attended six or more clinic visits compared to non-AHD clients (63%). The median number of visits was also comparable between the two groups. However, it is important to acknowledge that our analysis focused HIV care clinic visits. The dataset did not capture other forms of healthcare utilization that may disproportionately affect AHD clients such as hospitalizations, outpatient department (OPD) visits, oncology services, or other specialized care. As a result, while HIV clinic visit frequency appears similar, the broader resource burden on the health system may be underestimated for clients with AHD. [4].

While requiring more frequent clinic visits is understandable for patients with opportunistic infections and other symptoms of AHD [4], it may be a double-edged sword for ART outcomes. Clients who face barriers to visiting the clinic, costs, stigma, time, or other, are both more likely to develop AHD through late presentation of treatment and more likely to disengage from care if multiple visits are required. To address these challenges tailored service delivery models, such as multi-month dispensing and community ART groups, home delivery of ART, mobile clinics, workplace-based services, and fast-track refills may help reduce barriers to clinic visits and improve access to care, particularly for older, employed men who often present for ART initiation with advanced HIV disease [9,25].Differentiation between AHD clients who are actively ill and need acute clinical care and those with low CD4 counts but few symptoms of illness may be an important step in improving AHD service delivery.

Our findings further underscore this need by revealing differences in service preferences and utilization depending on how AHD was defined. Clients identified solely based on CD4 count <200 cells/μL (and who were asymptomatic) tended to prefer more frequent, clinic-based services, including monthly visits and appointments early in the month possibly

reflecting a heightened perceived need for monitoring. In contrast, clients who were both symptomatic (WHO stage 3/4) and had low CD4 counts were more likely to prefer external medication pick-up, suggesting limited mobility or a desire to minimize time spent in healthcare facilities due to illness severity (S4 and S6 Tables)

In addition to these clinical distinctions, sex also emerged as a key factor influencing service preferences and engagement. Our findings highlight important sex-based differences in both clinical care processes and service preferences among AHD clients. Female clients were more likely to have a documented six-month viral load test, suggesting either better retention in care or stronger alignment with routine monitoring protocols compared to their male counterparts. In contrast, male clients showed a stronger preference for individualized, one-on-one counselling, which may reflect a need for more tailored communication strategies to support engagement. Interestingly, while more males rated facility care as "as good as expected," a greater proportion of females rated it as "better than expected," suggesting differing perceptions of quality or expectations of care. Preferences for early morning appointments were more common among female clients, possibly reflecting caregiving responsibilities or employment constraints that make early access more desirable. These patterns underscore the value of offering flexible and differentiated service options, such as early clinic hours and individualized counselling, to better meet the diverse needs of AHD clients.

As noted, 15% (160/1098) of clients did not have a baseline CD4 count or WHO stage documented, and 34% (314/938) lacked a 6-month viral load test record. The absence of this information has significant implications, as it hinders accurate risk stratification, timely detection of treatment failure, and the ability to monitor patient response to antiretroviral therapy (ART) (S1 and S2 Tables). South African treatment guidelines rely on CD4 count to guide prophylactic and diagnostic steps, such as reflex testing for opportunistic infections like cryptococcosis or tuberculosis. CD4 count is also vital for baseline risk stratification, determining appropriate treatment and identifying patients at higher risk for complications and AHD. Viral load tests are essential for monitoring ART response, diagnosis of treatment failure or disease progression [26,27], and for determining eligibility for lower-intensity differentiated models of care. Addressing reasons for incomplete coverage of baseline CD4, staging, and viral load documentation may offer an opportunity to improve service delivery.

This study had several limitations. First, the exclusion of participants without documented baseline CD4 counts or WHO staging may have introduced selection bias, if those who were excluded were more likely to present with AHD or experience one of the study outcomes we investigated. The missing CD4 count and viral load (VL) baseline data can be attributed to several factors. For CD4 count, testing is typically done at ART initiation, but for clients re-engaging in care after a short treatment interruption, a repeat CD4 test may not always be conducted immediately, especially if the client is assessed clinically. For VL testing, it is scheduled at 3 or 6 months after ART initiation depending on which clinical guidelines are followed at the facility, and missing data may occur if clients are not retained in care long enough to reach this time point. Moreover, data linkage issues, such as duplicate files or mismatches between records, can also lead to missing results. These factors highlight the real-world challenges of HIV care, contributing to gaps in data that were beyond the scope of this study but are important to consider in future research.

Given that the characteristics of excluded individuals were similar to those included in the analysis (S1 Table), it seems unlikely that this potential source of bias had an important impact on the results. Second, viral load data were available for only a subset of participants (S6 Table) which may limit the generalizability of our findings regarding viral suppression. Third, although we observed few deaths in our study, this is likely due to incomplete documentation, which may have led to underreporting of mortality. Some clients who died could have been misclassified as disengaged from care due to lack of accurate death records [21]. Finally, the reliance on routine medical records and self-reported data for some variables could have introduced misclassification or recall bias.

## Conclusion

Despite the progress made in scaling up ART in South Africa, AHD remains a significant barrier to achieving optimal HIV care outcomes. AHD clients in our study experienced higher levels of low-level viremia within the first 6-months on ART

compared to non-AHD clients, reflecting the clinical complexity of AHD and the need for targeted interventions for this group. The persistence of AHD among ART initiators and the associated poor outcomes emphasize the need for differentiated care models and guidelines tailored to this vulnerable group. Differentiated care models and complementary interventions can further strengthen early HIV diagnosis, facilitate rapid ART initiation, support the management of comorbidities, and enhance adherence and retention in care. Expanding and targeting HIV testing through community-based and provider-initiated approaches can improve early case detection, particularly among underserved populations [28]. Enabling same-day ART initiation by improving facility readiness and using simplified regimens significantly reduces treatment delays and improves early viral suppression and retention outcomes [9,14]. Integrating care for opportunistic infections such as tuberculosis and cryptococcal meningitis, as well as non-communicable diseases, allows for more comprehensive care for individuals with complex clinical needs [29]. The use of point-of-care diagnostics, such as rapid CD4, CrAg, and TB-LAM tests facilitates timely identification and management of clients with advanced HIV disease (AHD). Digital adherence tools (e.g., SMS reminders, mobile applications) and peer support models have shown promise in improving adherence and engagement in care [30]. Multi-month dispensing, community ART groups, and fast-track refill systems, improve retention and patient satisfaction, especially among working-age men who are disproportionately affected by AHD [31,32]. Although many of these interventions are in place to varying degrees, strengthening the health system, through training, supportive supervision, and robust health information systems is essential to ensure consistent implementation and improved clinical outcomes.

## Supporting information

**S1 Table  Socio-demographic characteristics of PREFER study participants with or without baseline CD4 count.**
(DOCX)

**S2 Table  Preferences and expectations among AHD clients, stratified by AHD definition (CD4<200 cells/µL or WHO stage 3/4 conditions vs. CD4<200 cells/µL only).**
(DOCX)

**S3 Table  Service delivery preferences among AHD clients, stratified by sex.**
(DOCX)

**S4 Table  Healthcare resource use among AHD clients, stratified by AHD definition (CD4<200 cells/µL or WHO stage 3/4 conditions vs. CD4<200 cells/µL only).**
(DOCX)

**S5 Table  Healthcare resource use among AHD clients, stratified by sex.**
(DOCX)

**S6 Table  Socio-demographic characteristics of HIV clients stratified by six-month viral load documentation.**
(DOCX)

## Acknowledgments

The authors would like to acknowledge the contribution of the Retain6 study team and the participating healthcare facilities and study participants.

## Author contributions

**Conceptualization:** Elizabeth Kachingwe, Mariet Benade, Sydney Rosen, Mhairi Maskew.
**Data curation:** Nyasha Mutanda, Vinolia Ntjikelane, Mhairi Maskew.

**Formal analysis:** Elizabeth Kachingwe, Mhairi Maskew.

**Funding acquisition:** Sydney Rosen.

**Investigation:** Sydney Rosen.

**Methodology:** Elizabeth Kachingwe.

**Project administration:** Vinolia Ntjikelane.

**Supervision:** Sydney Rosen.

**Writing – original draft:** Elizabeth Kachingwe.

**Writing – review & editing:** Nyasha Mutanda, Vinolia Ntjikelane, Mariet Benade, Musa Manganye, Lufuno Malala, Sydney Rosen, Mhairi Maskew.

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
