## [Decision Letter · Decision Letter 0]

13 May 2025

PGPH-D-25-00852

Characteristics and six-month viral load suppression of clients presenting with advanced HIV disease in South Africa

Dear Dr. Elizabeth Kachingwe

Thank you for submitting your manuscript to PLOS Global Public Health. After careful consideration, we feel that it has merit but does not fully meet PLOS Global Public Health’s publication criteria as it currently stands. Therefore, we invite you to submit a revised version of the manuscript that addresses the points raised during the review process.

We look forward to receiving your revised manuscript.

Kind regards,

Sizulu Moyo, MBCBH, MPH, PhD

Academic Editor

Additional Editor Comments (if provided):

Reviewers' comments:

Reviewer's Responses to Questions

**Comments to the Author**

1. Does this manuscript meet PLOS Global Public Health’s publication criteria?

Reviewer #1: Yes

Reviewer #2: Yes

2. Has the statistical analysis been performed appropriately and rigorously?

Reviewer #1: Yes

Reviewer #2: Yes

3. Have the authors made all data underlying the findings in their manuscript fully available (please refer to the Data Availability Statement at the start of the manuscript PDF file)?

Reviewer #1: Yes

Reviewer #2: No

4. Is the manuscript presented in an intelligible fashion and written in standard English?

Reviewer #1: Yes

Reviewer #2: Yes

Reviewer #1: Many thanks for the opportunity to review this paper. I believe that it describes important research that aims to improve outcomes for those with AHD. My comments to the paper are as follows:

Introduction

Line 81-83 – perhaps rephrase or clarify why medication adherence and viral suppression is challenging for people with AHD compared to other people with HIV? Apart from potentially having more complex treatment regimens (presumably due to treatment of opportunistic infections), it is not clear why having AHD makes these goals harder?

Methods

It may be worth mentioned why data from Zambia was not used in this study.

How was re-initiation of ART defined? In Table 1 your refer to self-reported re-engagment in care after a period of disengagement or interruption – was there any time period specified for being out of care (either minimum, maximum or both)?

Results

It would be useful to show the number with measured CD4 and measured WHO stage. Your text mentioned that only 3% met both criteria for AHD (CD4 and WHO stage), but this may be due to low ascertainment of WHO stage? This relates to the authors final point in the conclusion regarding early detection of AHD, and the data may be relevant for regions where access to CD4 testing is especially limited.

Table 2 shows “How often client comes to this clinic for services for additional chronic services”. Please rephrase this to clarify. Is it possible to clarify if these additional chronic services are at separate appointments with other healthcare workers or would the same clinician at the facility see the client for HIV and their comorbidities at the same visit? The question is a little ambiguous as it does not necessarily mean that clients who come for additional chronic services attend the clinic more frequently, so it would be good to clarify this as this is one of your main points in the discussion section.

Table 2 – make it clear that the proportion of clients that have been treated for TB is calculated using the number ever diagnosed with TB

Do you have data on which ART regimen participants were receiving, and if not specifically for participants could you describe what proportion of clients at facilities are on DTG-based ART in the methods section, presumably most of them, given the timeframe of the study? This may be relevant as DTG-based ART can lead to more rapid viral suppression so those with low-level viraemia could in theory be those on non-INSTI based ART.

Among those with AHD, were the differences in preferences or resource utilization by sex?

Discussion

Line 261 – The authors state that “the higher frequency of healthcare visits for additional non-HIV services and prior TB diagnoses among AHD clients reflects the severe burden of comorbid conditions in this group…”. As mentioned above, I think this needs further clarification. Your results actually show a similar number of visits for AHD and non-AHD clients (table 5). It is not clear whether AHD clients are attending facilities with the same frequency but use more services during their visits, or whether they visit the clinic more frequently. The authors also mention higher frequency of prior TB diagnoses. The results (Table 2) do not report prior TB diagnoses but rather ever being diagnosed with TB, which would also include current TB. Since having TB is an indicator for AHD it would be necessary to present results which exclude current TB diagnoses for this statement to be justified.

Line 301-302: Suggest rephrase – The sentence starts off suggestion no difference, but ends stating that the difference was not statistically significant.

lines 298-309: This paragraph is unclear. It starts off suggesting that there was no difference between AHD and non-AHD clients’ visit frequency. There is then a suggestion that those with AHD would have higher visit frequency due to their condition and treatment needs. This statement does not mention the comorbidities and the extra services that AHD clients utilized for these that are described earlier in the discussion section, but only mentions the HIV-related OIs and symptoms of AHD as reasons for more frequent visits. I agree with the statements regarding the barriers to more frequent clinic visits especially for the potentially more vulnerable AHD clients, but your results are unclear whether AHD clients do in fact have more frequent visits. The authors suggest differentiating between clients who are ill and those who are asymptomatic as a way to improve AHD service delivery. To support this statement, would it be possible to disaggregate your results between these 2 groups for service preferences and utilization?

Reviewer #2: Characteristics and six-month viral load suppression of clients presenting with advanced HIV disease in South Africa

Overall Comments

Well-written article contributing key insights on patient characteristics and six-month viral suppression with a focus on those presenting comparison in clinical outcomes, care engagement, and service preferences between patients with advanced HIV disease and those without. I have a few comments on this article. Please see the sectional comments below

Title: The topic is well-suited for the journal.

Abstract:

Methods: Include the study design. Was it an observational cohort, etc.?

Conclusion: The authors only mention guideline updates to address ADH needs. How about other findings, such as implications of low viremia and service preferences?

Introduction

Lines 70-71-No need for parentheses

Lines 77-78: Was there a comparison group in the Sub-Saharan African study?

The last paragraph describes the focus of the paper. It could read better if you also include the unique contribution of your analysis compared to other studies.

Methods

What informed the categorization of the VL suppression categories? Is it based on the guidelines, and why were these thresholds chosen? Include the reason.

Discussion and conclusion

Lines 272-282: The author acknowledges a higher rate of low viremia but does not discuss the clinical and policy implications for managing low viremia

Lines 291-297: How many patients were likely to report hypertension? Any implications?

Lines 298-309: What can be done about clients who face barriers to clinic visits?

Limitations: The authors acknowledge missing CD4 and VL baseline data; however, the root cause is not mentioned.

Conclusion: Provide details on the targeted interventions and differentiated care models, as these are more general.

**Do you want your identity to be public for this peer review?** For information about this choice, including consent withdrawal, please see our Privacy Policy

Reviewer #1: No

Reviewer #2: No

---

## [Decision Letter · Decision Letter 1]

27 Aug 2025

Characteristics and six-month viral load suppression of clients presenting with advanced HIV disease in South Africa

PGPH-D-25-00852R1

Dear Elizabeth Kachingwe,

We are pleased to inform you that your manuscript 'Characteristics and six-month viral load suppression of clients presenting with advanced HIV disease in South Africa' has been provisionally accepted for publication in PLOS Global Public Health.

Best regards,

Sizulu Moyo, MBCHB, MPH, PhD

Academic Editor

Thank you addressing the comments. Please review for outstanding typographical errors. Review listing pregnancy as a comorbid condition because it is not. Yoi can remove it from Table 2 and report on it in the text. Secondly add if there was DR TB and the split between the AHD and non-AHD groups.

Reviewer Comments (if any, and for reference):

Reviewer's Responses to Questions

**Comments to the Author**

Reviewer #1: All comments have been addressed

Reviewer #2: All comments have been addressed

publication criteria?

Reviewer #1: Yes

Reviewer #2: Yes

3. Has the statistical analysis been performed appropriately and rigorously?

Reviewer #1: Yes

Reviewer #2: Yes

4. Have the authors made all data underlying the findings in their manuscript fully available (please refer to the Data Availability Statement at the start of the manuscript PDF file)?

Reviewer #1: No

Reviewer #2: Yes

5. Is the manuscript presented in an intelligible fashion and written in standard English?

Reviewer #1: Yes

Reviewer #2: Yes

Reviewer #1: Thanks for the very thorough response to my comments.

One minor suggestion: In the discussion you have added a paragraph looking at sex-based differences in service preferences. It may be more appropriate to use "sex" in this paragraph rather than "gender", unless you specifically had data on gender.

Reviewer #2: Thank you for the opportunity to review this important work, which advances HIV care and management. All my prior comments have been addressed, and I have no further feedback. I look forward to seeing this work published.

**Do you want your identity to be public for this peer review?** For information about this choice, including consent withdrawal, please see our Privacy Policy

Reviewer #1: No

Reviewer #2: No
